# Resistance to thyroid hormone induced tachycardia in RTHα syndrome

Riccardo Dore [1], Laura Watson[2], Stefanie Hollidge[3], Christin Krause[4], Sarah Christine Sentis[1], Rebecca Oelkrug[1], Cathleen Geißler[1], Kornelia Johann [1], Mehdi Pedaran[1], Greta Lyons[5], Nuria Lopez-Alcantara[1], Julia Resch[1], Friedhelm Sayk[6], Karl Alexander Iwen[1,6], Andre Franke [7], Teide Jens Boysen[7], Jeffrey W. Dalley [8,9], Kristina Lorenz[10,11], Carla Moran[5,12], Kirsten L. Rennie[3], Anders Arner[13], Henriette Kirchner[4], Krishna Chatterjee [5] & Jens Mittag [1] ✉

Mutations in thyroid hormone receptor α1 (TRα1) cause Resistance to Thyroid Hormone α (RTHα), a disorder characterized by hypothyroidism in TRα1-expressing tissues including the heart. Surprisingly, we report that treatment of RTHα patients with thyroxine to overcome tissue hormone resistance does not elevate their heart rate. Cardiac telemetry in male, TRα1 mutant, mice indicates that such persistent bradycardia is caused by an intrinsic cardiac defect and not due to altered autonomic control. Transcriptomic analyses show preserved, thyroid hormone (T3)-dependent upregulation of pacemaker channels (*Hcn2, Hcn4*), but irreversibly reduced expression of several ion channel genes controlling heart rate. Exposure of TRα1 mutant male mice to higher maternal T3 concentrations in utero, restores altered expression and DNA methylation of ion channels, including *Ryr2*. Our findings indicate that target genes other than *Hcn2* and *Hcn4* mediate T3-induced tachycardia and suggest that treatment of RTHα patients with thyroxine in high dosage without concomitant tachycardia, is possible.

Thyroid hormones (THs) exert profound chronotropic and inotropic cardiovascular effects, manifesting as bradycardia in hypothyroidism whilst tachycardia is a leading symptom of hyperthyroidism[1,2]. The actions of THs are mediated by their nuclear TH receptors, TRα1 and TRβ, which are encoded by the two different genes *THRA* and *THRB*. In the heart, TRα1 is the predominant isoform and mediates the known effects of the hormone such as increasing contractility, oxygen consumption, cardiac output as well as heart rate[2–4]. Several bona fide target genes of TH action in the heart have been identified, including the pacemaker channels HCN2 and HCN4 (Hyperpolarization-activated Cyclic Nucleotide-gated Channel 2 and 4), which conduct the "funny current" in the sinoatrial node that generates slow diastolic

[1]Institute for Endocrinology and Diabetes, Center of Brain Behavior & Metabolism, University of Lübeck, Ratzeburger Allee 160, 23562 Lübeck, Germany. [2]National Institute Health and Care Research Cambridge Clinical Research Facility, Addenbrooke's Hospital, Cambridge, UK. [3]MRC Epidemiology Unit and Wellcome-MRC Institute of Metabolic Science, University of Cambridge, Cambridge, UK. [4]Institute for Human Genetics, Department of Epigenetics & Metabolism, Center of Brain Behavior & Metabolism, University of Lübeck, Ratzeburger Allee 160, 23562 Lübeck, Germany. [5]Wellcome-MRC Institute of Metabolic Science, Metabolic Research Laboratories, University of Cambridge, Cambridge, UK. [6]Internal Medicine I, Universitätsklinikum Schleswig-Holstein, Ratzeburger Allee 160, 23562 Lübeck, Germany. [7]Institute of Clinical Molecular Biology, Christian-Albrechts-University of Kiel, Rosalind-Franklin-Straße 12, 24105 Kiel, Germany. [8]Department of Psychology, University of Cambridge, Cambridge CB2 3EB, UK. [9]Department of Psychiatry, University of Cambridge, Cambridge CB2 2QQ, UK. [10]Institute of Pharmacology and Toxicology, University of Würzburg, Versbacher Straße 9, 97078 Wuerzburg, Germany. [11]Leibniz-Institut für Analytische Wissenschaften-ISAS-e.V., Bunsen-Kirchhoff-Str. 11, 44139 Dortmund, Germany. [12]Beacon Hospital and School of Medicine, University College Dublin, Dublin, Ireland. [13]Department of Clinical Sciences, Lund University, c/o Igelösa Life Science AB, Igelösa 373, 225 94, Lund, Sweden. ✉e-mail: jens.mittag@uni-luebeck.de

depolarization to the threshold, initiating a propagated action potential spreading through the entire heart. Additionally, the voltage-gated potassium channels KCNA5 and KCND2 that contribute to the early repolarization of cardiomyocytes as well as the calcium flux modulating proteins ATP2A2 (SERCA2) and RYR2 are also known targets of TH[3,5,6]. In addition to the direct actions on peripheral tissues, TH can also act centrally in the brain, altering the output of the autonomic nervous system to liver or brown adipose tissue[7,8]. Although this paradigm of acute central action of TH has not yet been convincingly demonstrated for the cardiovascular system[9], it is widely assumed that hyperthyroidism reduces the vagal tone of the parasympathetic nervous system (PSNS) while activating the sympathetic nervous system (SNS). Together with proposed increases in activity of the adrenergic receptor ADRB1 and the pacemaker channels HCN2 and HCN4, these mechanisms are currently assumed to mediate the tachycardia of thyrotoxicosis[2].

Studies of TRα1 knockout mice, which exhibit bradycardia, underscore the important role of this receptor isoform in controlling heart rate[10]. Subsequently, insights into the interplay between central and peripheral actions of TH were gained using mice heterozygous for the R384C mutation in TRα1 (TRα1 + m), which better resembles the situation found in the respective patients. This unique animal model has the particular advantage that this mutation only partially reduces 3,3',5-triiodothyronine (T3) binding to the receptor, such that it inhibits the function of its wild-type counterpart in a dominant negative manner in the presence of physiological TH concentrations; however, treating these mice with T3 in pharmacological dosage reactivates TRα1 signalling fully in vivo[11,12]. Accordingly, TRα1 + m mice enable differentiation of the acute versus permanent effects of defective, mutant TRα1 during mouse development[13,14].

The human disorder of Resistance to Thyroid Hormone α (RTHα), or congenital, nongoitrous hypothyroidism 6 (OMIM 614450), due to heterozygous mutations in TRα1, is characterized by refractoriness to TH action in TRα1-expressing tissues (e.g., central nervous system, bone, myocardium, skeletal muscle, gastrointestinal tract)[15,16]. Thyroxine therapy of RTHα is beneficial, improving linear growth, increasing resting energy expenditure to limit weight gain, alleviating constipation and enhancing wellbeing and affect of patients[17–19]. However, as treatment with high dosages of thyroxine is required to overcome tissue hormone resistance, this therapeutic approach could potentially be associated with thyrotoxic side effects including in the cardiovascular system.

In this work, we unexpectedly find that high dose thyroxine therapy of RTHα patients with a partial loss-of-function mutation (A263V) in TRα[18] does not elevate their heart rate. Investigating underlying mechanisms mediating this, we studied the cardiac phenotype in mice harboring a TRα1R384C mutation, which exhibits similar, partial, loss-of-function. Using these mice, we find that the effects of TH on heart rate are largely direct, whilst its effects via the autonomic nervous system predominantly affect heart rate frequency distribution rather than average heart rate. Combining cardiac telemetry and microarray-based transcriptomics, we demonstrate that the TH mediated upregulation of *Hcn2* and *Hcn4* expression is not associated with changes in heart rate. Furthermore, we find that impaired TRα1 action during mouse development programs irreversible changes in expression of multiple cardiac genes, including potassium and calcium channels. We suggest that such developmental programming by altered epigenetic DNA methylation accounts for the observed lack of elevation in heart rate following treatment of RTHα patients with thyroxine in high dosage.

## Results

### Thyroxine therapy of RTHα patients in high dosage, does not elevate heart rate

Three RTHα patients, harboring a mutation (TRα1A263V) whose loss-of-function can be overcome as demonstrated in previous in vitro studies[18], were treated with thyroxine in supraphysiologic dosage, leading to suppressed circulating TSH and raised free T3 concentrations (Fig. 1a and Supplementary Fig. 1A). Although such therapy overcame tissue hormone resistance, as evidenced by elevation of resting energy expenditure or metabolic rate in patients from markedly subnormal to higher levels (Supplementary Fig. 1A), their average heart rate did not increase (Fig. 1b and Supplementary Fig. 1A) and was significantly lower than in patients with conventional thyrotoxicosis (Fig. 1b, Supplementary Fig. 1A, and Supplementary Table 1), suggesting possible resistance to TH-induced tachycardia in this disorder.

### Heart rate in mice with impaired TRα1 signalling

To identify a possible mechanism for this resistance, we studied mice heterozygous for the TRα1R384C mutation, an established animal model for RTHα. To enable best possible comparison with the human disorder, we used a wireless radiotelemetry system in conscious and freely moving animals, thus avoiding possible interference from effects of anesthesia or handling stress. In addition to experiments at room temperature (22 °C), we also undertook recordings at thermoneutrality (30 °C) when autonomic tone in mice resembles that of humans more closely, with prominent parasympathetic nervous system (PSNS) and reduced sympathetic nervous system (SNS) inputs[20]. Moreover, at this higher temperature the animals are no longer cold-stressed and exhibit a more normal metabolic rate[21].

At both temperatures, we observed lower heart rate in TRα1 + m mice as compared to controls, both in the active as well as in the resting light phase (Fig. 1c–f). TRα1 + m mutants also exhibited reduced locomotor activity at both temperatures (Supplementary Fig. 1B, C). The heart rate frequency distribution of the mutants was broader at 22 °C but narrower at 30 °C as compared to wild-types with a lower average heart rate at both temperatures (Fig. 1e, f). When we tested autonomic activity using pharmacological blockade of PSNS (methylscopolamine) and SNS (timolol maleate) in vivo, we observed a shift from predominantly SNS at 22 °C to more PSNS control in wild-types at 30 °C (Fig. 1g, h), as expected from previous studies[11,20]. TRα1 + m mice had reduced PSNS and SNS activity at 22 °C, which however was no longer different at 30 °C (Fig. 1g, h). Together with the lower intrinsic heart rate after complete pharmacological autonomic receptor blockade in TRα1 + m mice at both temperatures (Fig. 1g, h), these observations indicate that TH act directly on the heart via TRα1, to control heart rate. Given that SNS/PSNS activity of mice resembles humans more closely at 30 °C, with negligible defects in the autonomic control of heart rate in TRα1+m animals, subsequent experiments were all conducted at this temperature.

### Reactivation of TRα1 signalling in vivo

Knowing that defective TRα1 signalling can be restored by elevating T3 levels in TRα1 + m mice, we treated wild-type and TRα1 + m mice with T3 in the drinking water for 12 days at 30 °C, resulting in ~sixfold elevation in serum T3 and strongly suppressed T4 concentrations in both groups, with somewhat higher absolute levels in TRα1 + m mice (Supplementary Fig. 1D, E). Most remarkably, whilst such treatment induced the expected tachycardia in wild-type mice, TRα1 + m mice remained bradycardic (Fig. 2a). In both genotypes, locomotion was generally higher following T3 treatment, with activity of TRα1 + m mutant animals no longer being significantly different during the night (Supplementary Fig. 1F). Furthermore, T3 treatment reduced PSNS activity in both genotypes, which should facilitate the development of tachycardia (Fig. 2b), while SNS activity was not significantly affected by T3 (Fig. 2b), concordant with unchanged norepinephrine turnover in the hearts of T3-treated animals (Supplementary Fig. 2A). After complete, pharmacological, autonomic receptors blockade, heart rate remained much lower in TRα1 + m mice (Fig. 2c), indicating that resistance to T3-induced tachycardia is due to an intrinsic cardiac

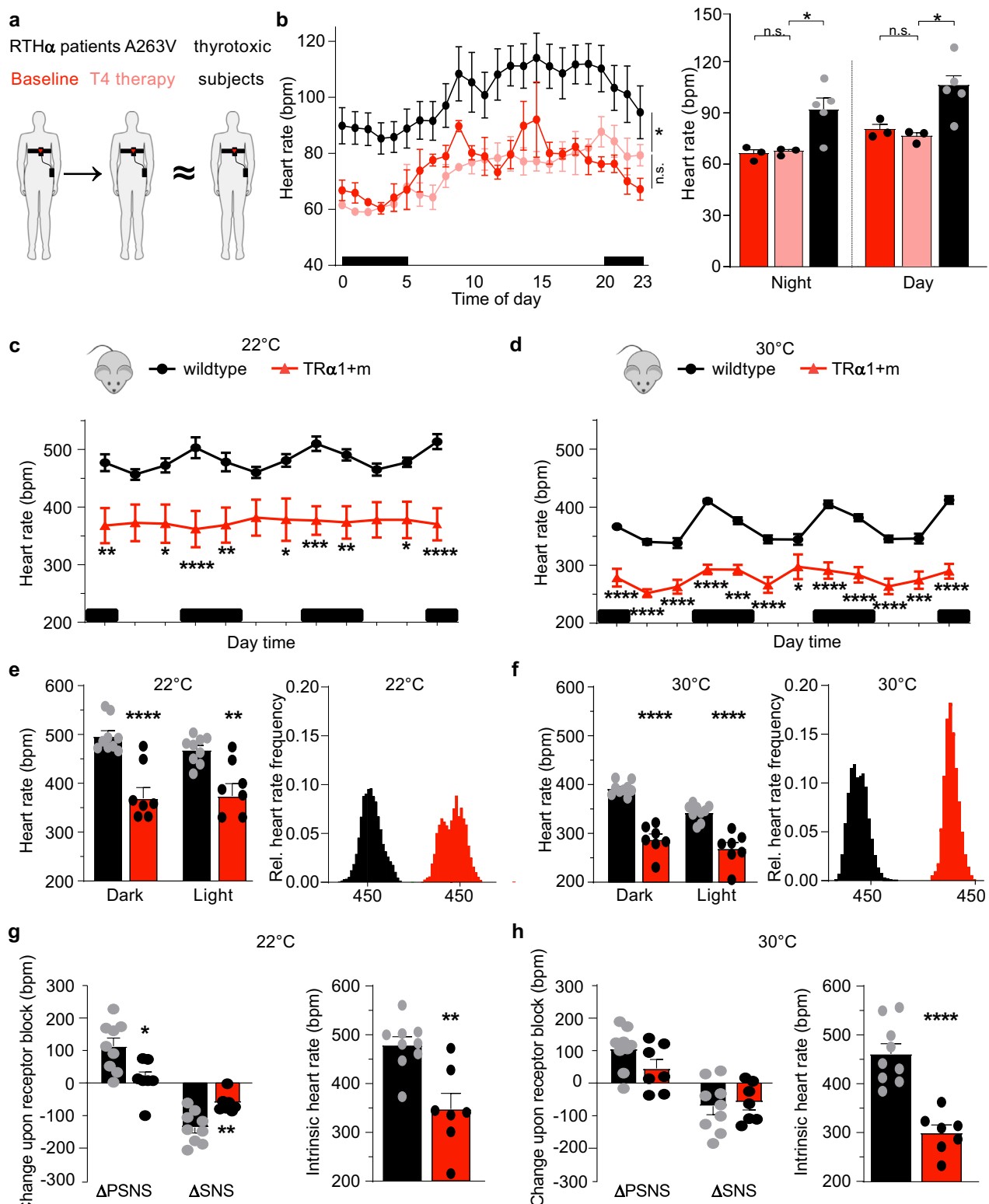

**Fig. 1 | Heart Rate in RTHα individuals and mice at 22 °C and 30 °C. a** 24-h heart rate profiles were recorded in RTHα patients at baseline (red) or following thyroxine (T4) therapy (light red) or in patients with thyrotoxicosis (black). **b** Heart rate profiles in RTHα patients (*n* = 3) at baseline or following T4 therapy or in patients with thyrotoxicosis (*n* = 5) over 24 h (left panel) or divided into Day and Night time (right panel). **c** Radiotelemetry recordings of heart rate in TRα1 + m (red) and wild-type controls (black) over 3 days at 22 °C or **d** at 30 °C. Black bars indicate dark time activity period. **e** Average heart rate for dark or light phase and heart rate distribution of these animals at 22 °C or **f** at 30 °C. **g** Contributions of the para-sympathetic (PSNS) or sympathetic nervous system (SNS) in these animals as determined by change in heart rate upon pharmacological blockade with methylscopolamine or timolol as well as intrinsic heart rate after full pharmaco-logical blockade with these drugs at 22 °C or **h** at 30 °C. Values are mean ± SEM for *n* = 3–5 individuals per group (**a**, **b**, paired comparison for RTHα patients before and after thyroxine therapy, non-paired comparison for RTHα patients after thyr-oxine therapy and thyrotoxic subjects) or *n* = 9 wild-type controls and *n* = 7 TRα1 + m mutants (**c**–**h**). \**P* < 0.05; \*\**P* < 0.01; \*\*\**P* < 0.001 and \*\*\*\**P* < 0.0001 with two-way ANOVA and Sidak's multiple comparison tests or two-tailed Student´s *t* test. Exact *P* values are provided in Supplementary Table 1.

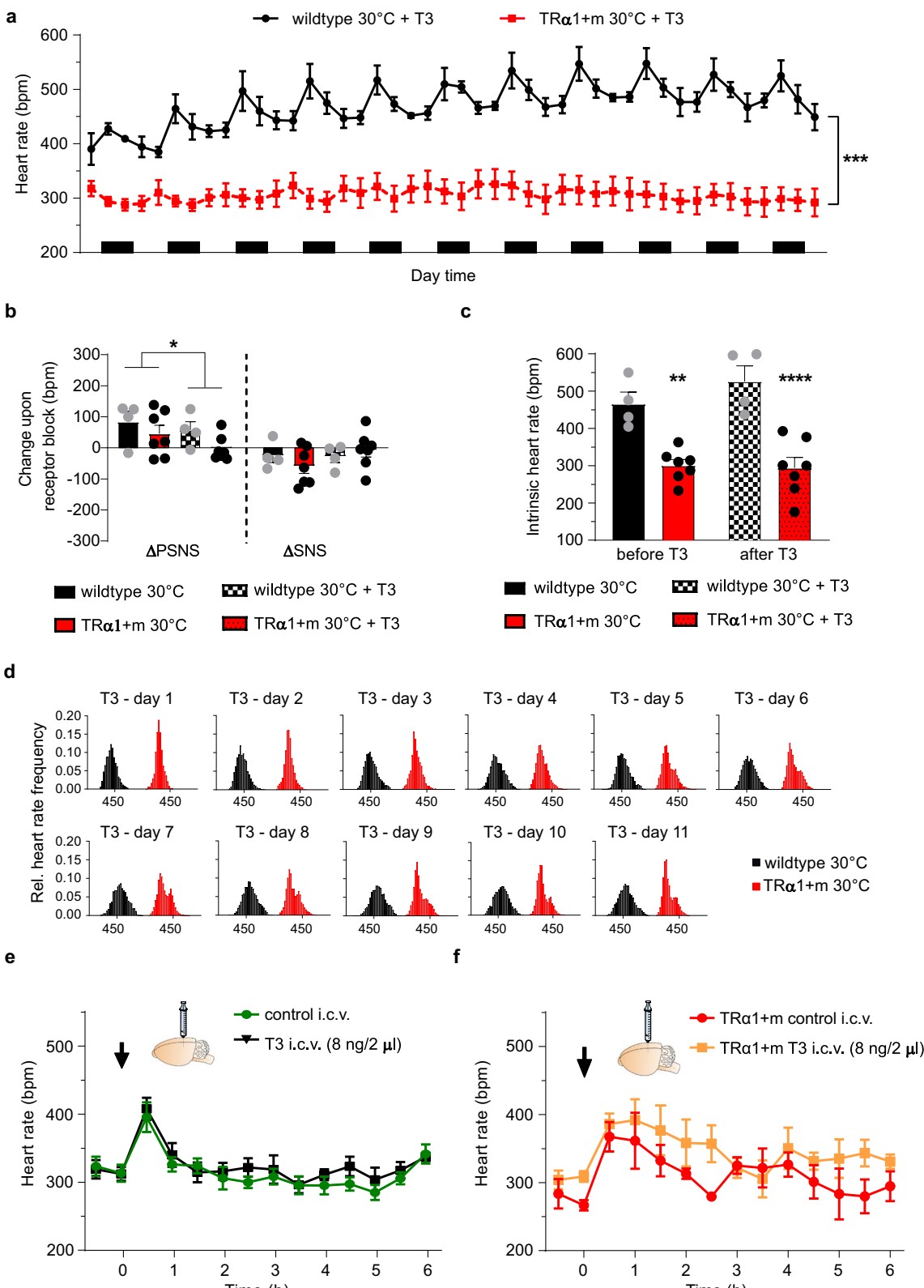

**Fig. 2 | Reactivation of the mutant TRα1 with T3 and central T3 i.c.v. administration at 30 °C. a** Radiotelemetry recordings of heart rate in TRα1 + m (red) as well as wild-type controls (black) over 12 days of oral T3 treatment at 30 °C. Black bars indicate dark time activity period. **b** Contributions of the parasympathetic (PSNS) or sympathetic nervous system (SNS) in these animals as determined by change in heart rate upon pharmacological blockade with methylscopolamine or timolol at 30 °C before and after 12 days of T3 treatment. **c** Intrinsic heart rate after full pharmacological blockade in these animals at 30 °C before and after 12 days of

T3 treatment. **d** Development of heart rate distribution over 11 consecutive days of T3 treatment at 30 °C in these animals. **e** Heart rate in wild-type or TRα1 + m (**f**) animals after i.c.v. injection of 8 ng T3 (black or orange) or control solution (green or red) as recorded by radiotelemetry for 6 h at 30 °C. Values are mean ± SEM for n = 4 wild-type controls and n = 7 TRα1 + m mutants; n = 5 wild-types and n = 3 TRα1 + m mutants for i.c.v. study. *P < 0.05; **P < 0.01; ***P < 0.001; ****P < 0.0001 with two-way ANOVA and Sidak's multiple comparison tests. Exact P values are provided in Supplementary Table 1.

cause. However, the frequency distribution of heart rate broadened in both wild-type and TRα1+m mice during T3 treatment (Fig. 2d), possibly suggesting that it is less tightly regulated via the autonomic nervous system.

To test the contribution of central actions of TH to heart rate directly, we administered T3 intracerebroventricularly (i.c.v.) to wild-type and TRα1+m mice monitored with radiotelemetry. We observed no significant change in heart rate 6 h after T3 administration in either genotype (Fig. 2e, f), a time frame that is known to trigger central effects of TH on liver and brown adipose tissue[8,22]. Likewise, 6 h after i.c.v. administration of T3, no changes in serum T3 and T4 concentrations (Supplementary Fig. 2B), SNS or PSNS activity (Supplementary Fig. 2C), or locomotion were seen (Supplementary Fig. 2D, E).

### Irreversibly altered expression of T3-regulated genes governing cardiomyocyte action potential in TRα1 + m mice

To investigate the molecular basis for resistance to T3-induced tachycardia, we undertook microarray profiling of gene expression in hearts of wild-type and TRα1+m mice, acclimated at 30 °C, before and after T3 treatment. Principal component analysis showed distinct clustering of the four different groups (Fig. 3a). Whilst 240 genes were T3-regulated in wild-type animals, 161 genes were differentially expressed between wild-type and TRα1+m mice and 345 genes being differentially expressed between wild-type or TRα1+m mice after receptor reactivation by T3 (Fig. 3b). Cardiac target genes, known to be T3-regulated in mice at room temperature[2,3], were also downregulated (Myh7) or induced (Ryr2) at 30 °C (Fig. 3c and Supplementary Fig. 3A). In addition, basal expression of genes (Myh6, Ryr2, Atp2a2) was lower or higher (Myh7, Adrb1) in TRα1+m mice, with Chrm2 expression being unchanged (Fig. 3c and Supplementary Fig. 3A). Contrary to expectations[2], T3 treatment did not affect Adrb1 expression at 30 °C (Supplementary Fig. 3A), suggesting that upregulation of this receptor is unlikely to account for the known increased cardiac sympathetic hypersensitivity of hyperthyroidism. As expected, expression of the pacemaker gene Hcn4 was downregulated in untreated TRα1+m mice, with both Hcn2 and Hcn4 being strongly T3-induced in both genotypes[2,3], showing that T3 treatment of TRα1+m mice reactivates normal TRα1 signalling in the heart (Fig. 3c). Unbiased gene ontology pathway analyses revealed several T3-regulated pathways affected by T3 including factors involved in metabolic and immune processes and regulation of heart rate (Supplementary Fig. 3B, top ten genes for each factor in Supplementary Fig. 3C). Accordingly, we then focussed on genes involved in the regulation of heart rate (GO:0002027 plus potassium channels), revealing several T3-regulated bona fide TRα1 target genes (Fig. 3d) as well as genes that were regulated by T3 without a TRα1 contribution (Supplementary Fig. 3D). Most importantly, we identified several genes whose expression was irreversibly altered in TRα1+m mice (Fig. 3e), including potassium channel genes such as Kcnh2 as well as the calcium channel Ryr2, providing a possible molecular explanation for cardiac T3 resistance in these animals.

### Developmental effect of mutant TRα1 affects cardiac gene expression

To investigate whether the irreversibly altered expression of these genes was caused by developmental actions of mutant TRα1, we treated the mothers of TRα1+m animals with T3 in the second half of pregnancy (Fig. 4a), to reactivate TRα1 signalling specifically during this critical window of cardiac, embryonic, development[23]. As expected, the treatment led to highly elevated circulating total serum T3 with suppressed total T4 in the pregnant dams and significantly elevated serum T3 levels in the embryos at embryonal day 17 (Supplementary Fig. 4A). Following cessation of T3 treatment (one day before birth), wild-type and TRα1+m offspring were allowed to grow up without further intervention and analysed in adulthood, showing that the time-limited T3 treatment in utero had no effect on the thyroid

hormone levels when compared to untreated animals of the same genotype (Supplementary Fig. 4A, right panel). We then again analysed gene expression by microarray analysis in these animals: the principal component analysis revealed distinct clustering of the two genotypes that were clearly shifted by the maternal T3 treatment (Fig. 4b) with several genes regulated in each condition (Supplementary Fig. 4B). When we focussed on genes involved in the regulation of heart rate (GO:0002027 plus potassium channels), we found that T3 treatment normalized expression of a subset of them (Fig. 4c, top), whereas the expression of other genes was altered by maternal T3 exposure irrespective of the genotype of animals (Fig. 4c, bottom). Hypothesising that this altered gene expression landscape could be mediated by altered fetal programming via CpG DNA methylation, we undertook genome-wide methylation analyses using the Infinium Mouse Methylation BeadChip array. Principal component analysis showed distinct clustering of CpG DNA methylation according to genotype, again with a noticeable shift induced by maternal T3 treatment (Fig. 4d), consistent with an overall reduction in average DNA methylation in TRα1+m mice which was reversed by T3 treatment (Fig. 4e). However, the majority of CpG sites showed either little or almost complete methylation, with a similar methylation density pattern in all conditions (Fig. 4f and Supplementary Fig. 4C), suggesting that observed changes in overall average methylation are not driven by a genome-wide effect (i.e., entire density shift to the left or right), but rather a site-specific effect. Concurrently, when the differentially methylated CpG sites were unbiasedly assembled in a heat map, several clusters (Fig. 4g, arrowheads) representing CpG methylation patterns specifically altered in TRα1+m mice and normalized by maternal T3 treatment were observed, showing a strong link between embryonal TRα1 signalling and cardiac DNA methylation. Finally, when we analysed CpG methylation at the level of individual genes, we identified a specific CpG site within the body of Ryr2 gene whose methylation was increased in TRα1+m mutant mice, with reversal of this by maternal T3 treatment (Fig. 4h). For Kcnh2, we identified a CpG site whose methylation was increased by T3 exposure in both genotypes (Fig. 4i).

## Discussion

Current knowledge suggests that TH controls heart rate by regulating the expression of Hcn2 and Hcn4 pacemaker channel genes. Furthermore, given known similarities of features of hyperthyroidism or overactivity of the sympathetic nervous system, it is thought that TH controls heart rate by stimulation of the SNS. Prompted by clinical observations in patients with RTHα, our studies of TRα1+m mutant mice at thermoneutrality (30 °C), where animals are not cold-stressed[20,21] and, as in humans, heart rate is predominantly controlled by parasympathetic activity, challenge both these assumptions.

It is recognized that TH acts indirectly, affecting efferent SNS output from the central nervous system via TRα1, to regulate metabolic activity in tissues like liver and brown adipose tissue - often in synergy with the direct peripheral actions of the hormone[7,8,22]. Our observations suggest that this does not seem to be the case for heart rate, as we did not observe altered SNS activity in TRα1 mutant mice at thermoneutrality (30 °C) or following central, i.c.v., administration of T3. Furthermore, norepinephrine turnover, a marker of SNS activity in the heart, was not altered in T3-treated animals. Our findings are in good agreement with a previous study showing that the T3-induced tachycardia persisted in mice lacking all β-adrenergic receptors[24], and even earlier work showing no change in β-adrenergic responses in hyper- or hypothyroidism presumably due to intracardiac compensatory mechanisms[25]. However, it is important to note that several previous studies in mice were conducted at room temperature (22 °C), with expression regulation of some genes by TH, including Adrb1, differing in this context[3,26]. Coupled with the fact that the SNS is more active in mice at 22 °C, such complexities in the interplay between indirect SNS and direct cardiac actions of TH, may have led to an

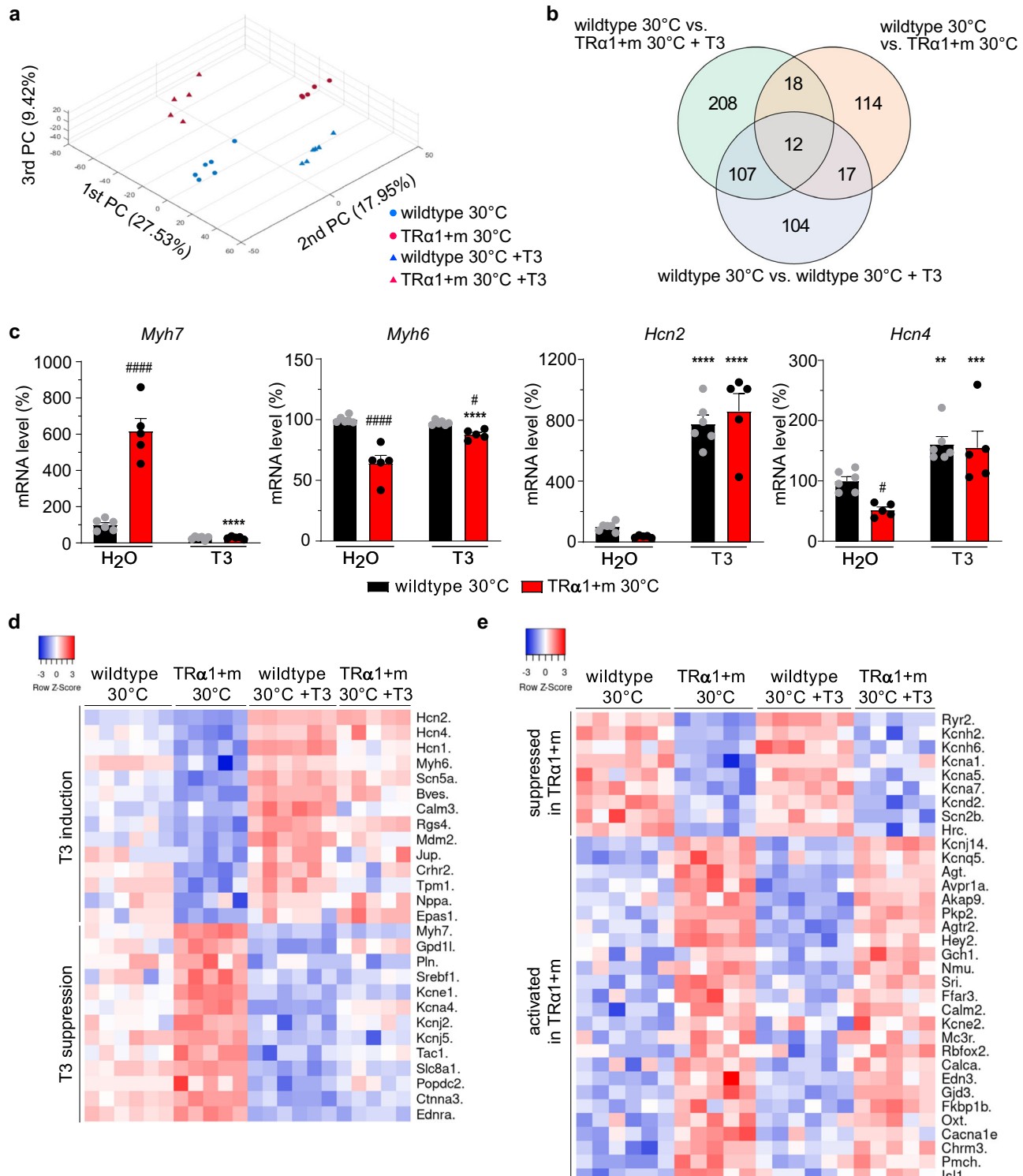

**Fig. 3 | Microarray gene expression analysis of T3 response in TRα1 + m and control animals at 30 °C. a** Three-dimensional principal component analysis in TRα1 + m (red) as well as wild-type controls (blue) at 30 °C with (triangles) or without T3 treatment (circles). **b** Venn-Diagram indicating the number of changed genes under the different conditions as indicated. **c** Expression of the known T3-responsive genes myosin-heavy-chain 7 (*Myh7*) and 6 (*Myh6*), as well as the hyperpolarization-activated cyclic nucleotide-gated channels 2 and 4 (*Hcn2, Hcn4*) in these animals (*n* = 6 wild-types in black, *n* = 5 TRα1 + m mutants in red). **d** Heat map for bona fide TRα1 target genes involved in heart rate regulation or potassium channels whose expression at baseline in wild-type and TRα1 + m mutants is

significantly different, with elimination of differences by T3 treatment). **e** Heat map for genes involved in heart rate regulation or potassium channels whose expression at baseline in wild-type and TRα1 + m mutants is significantly different, with inability of T3 treatment to reverse such differences in TRα1 + m mutants. Values are mean ± SEM for *n* = 5/6 per group. **P < 0.01; ***P < 0.001, and ****P < 0.0001 for T3 effect; #P < 0.05, and ####P < 0.0001 for TRα1 effect; two-way ANOVA with post hoc test corrected for multiple comparisons by controlling the false discovery rate using the Benjamini, Krieger and Yekutieli method. Exact *P* values are provided in Supplementary Table 1.

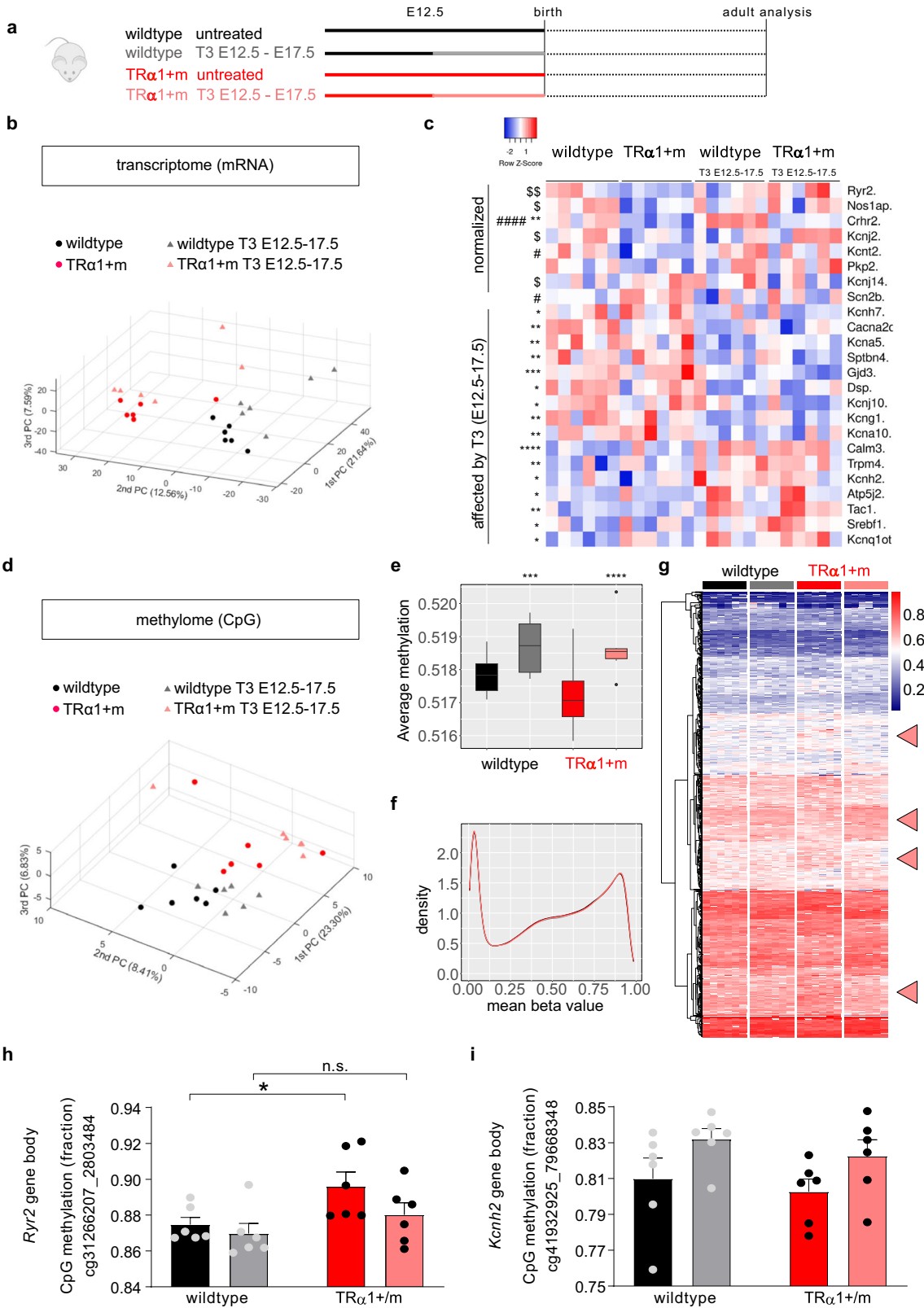

overestimation of the SNS contribution to specific phenotypes in animals studied at temperatures below thermoneutrality.

In contrast to the SNS, we observed a slight reduction in PSNS activity following T3 administration at 30 °C in both genotypes, reflecting lower vagal activity in hyperthyroidism as reported previously[27,28]. Although reduced PSNS activity should result in elevation of heart rate, this was not observed in TRα1 + m mice

suggesting that the hearts of these animals are less sensitive to altered PSNS activity. Furthermore, in denervation studies, TRα1 + m mice exhibited lower overall PSNS input—also a characteristic seen in hypothyroid patients[29]. Nevertheless, our observation that no change in SNS or PSNS activity was seen following i.c.v administration of T3 in wild-type mice, suggests that the interaction between TH and the PSNS control of heart rate may be largely mediated at the level of

**Fig. 4 | Developmental programming in TRα1 + m and wild-type animals by thyroid hormone in the second half of pregnancy. a** Treatment scheme of TRα1 + m mutants (red) or wild-type controls (black) that were untreated (solid color) or treated with T3 (light color) from embryonal day 12.5 (E12.5) to E17.5 by oral T3 treatment of the pregnant mother and analysed as adults. **b** Three-dimensional principal component analysis of the transcriptome microarray data of these animals. **c** Heat map for genes involved in heart rate regulation or potassium channels that are normalized in TRa1+m animals by maternal T3 treatment (upper part) or significantly affected by the maternal T3 treatment irrespective of genotype (lower part). **d** Three-dimensional principal component analysis of the methylome (CpG DNA methylation) data of these animals. **e** Average global DNA methylation in these groups. **f** Distribution of the relative DNA methylation ranging from fully demethylated to fully methylated CpG. **g** Unbiased clustering of differentially methylated CpG sites in the four groups. Arrowheads highlight clusters with noticeably changed methylation in TRα1 + m

mice that are reversed upon maternal treatment. **h** Increased DNA methylation at a specific CpG site within the body of the *Ryr2* gene in TRα1 + m mice as compared to wild-types, which is reversed by maternal T3 treatment. **i** Increased DNA methylation at a specific CpG site within the body of the *Kcnh2* gene in both wild-type and TRα1 + m mice upon maternal T3 treatment. Values are mean ± SEM for $n = 6$ per group. **c** $*P < 0.05$; $**P < 0.01$; $***P < 0.001$, and $****P < 0.0001$ with two-way ANOVA for T3 effect; $^{\#}P < 0.05$, and $^{\#\#\#\#}P < 0.0001$ with two-way ANOVA for TRα1 effect; $^{\$}P < 0.05$, and $^{\$\$}P < 0.01$ with two-way ANOVA for TRα1 and T3 interaction effect. **e** Data are shown as box plots (box represents median with interquartile range, ie. 25th to 75th centiles, whiskers depict range of values from minimum to maximum); $***P < 0.001$, and $****P < 0.0001$, Kruskal–Wallis with Wilcoxon rank-sum test with continuity correction. **h** $*P < 0.05$ for TRα1 effect, two-way ANOVA with post hoc test corrected for multiple comparisons by controlling the false discovery rate using the Benjamini, Krieger and Yekutieli method. Exact *P* values are provided in Supplementary Table 1.

cardiac signal transduction. Overall, our data suggest that the acute central autonomic effects of TH on average heart rate may be negligible, whilst T3 may still influence other cardiovascular parameters such as heart rate frequency distribution, as observed in an animal model of central hypothyroidism[30].

It has long been recognized that TH positively regulates expression of sinoatrial pacemaker channels *Hcn2* and *Hcn4*[1,3,6,31], providing a potential molecular mechanism for the chronotropic effects of the hormone. However, our observations in T3-treated TRα1 + m mice at 30 °C, dissociate changes in expression of these genes from heart rate: whilst *Hcn2* and *Hcn4* expression was strongly induced as expected[3,6], the animals remained bradycardic with no change in heart rate following T3 administration, indicating that neither *Hcn2* nor *Hcn4* were determinants of this. Our observations are supported by recent animal studies, showing that *Hcn2* knockout mice exhibit normal heart rate[32]. Likewise, many inducible, cardiac-specific *Hcn4* knockout models also showed normal heart rate, although these findings are somewhat controversial[33]. Accordingly, it is hypothesised that rather than controlling pacemaker activity, these *Hcn* channels function to stabilize heart rate and integrate autonomic signals, with increased sensitivity to PSNS activity seen in an inducible *Hcn4* knockout model supporting this notion[33,34]. It is therefore tempting to speculate that the observed T3 effects on *Hcn2* and/or *Hcn4* expression rather mediate altered PSNS responses seen in hyperthyroid animals.

At the molecular level, the effects of TH on duration of ventricular action potential have been associated with regulation of several potassium channels including *Kcnd2* or *Kcna5* by T3[35,36]. Likewise, thyroid hormone also affects cardiac contraction and relaxation speed by modulating calcium flux between cytosol and sarcoplasmic reticulum, altering *Atp2a2* and *Ryr2* gene expression[37]. However, our observations in TRα1 + m animals studied at thermoneutrality, indicate that expression of these genes (together with delayed rectifier channels e.g., *Kcnh2*) is irreversibly reduced and not fully T3 inducible. This indicates that the effects of TH on action potential duration are complex, with a combination of irreversible, developmentally programmed, effects and preserved acute responses to hormone (e.g., T3-mediated repression of *Kcne1* via TRα1[38]). Overall, our microarray expression profiling data, obtained in animals studied at thermoneutrality thereby obviating confounding effects of cold stress or SNS hyperactivity, provide a unique and comprehensive roadmap for future studies to dissect the effects of TH on cardiovascular function.

Based on the finding that the expression of several genes was permanently altered in TRα1 + m mice and not restored by receptor reactivation, we hypothesised that their expression had been developmentally reprogrammed by altered DNA methylation—an epigenetic effect of maternal TH that has been observed previously in other tissues[39]. Indeed, our observations indicate that several clusters of CpG sites are differentially methylated in TRα1 + m mice, with maternal T3 treatment normalising these differences. Combining our transcriptome

and methylome observations, *Ryr2* emerged as the most promising candidate, as its reduced expression in TRα1 + m mice is rescued by maternal T3 exposure but not by treating adult mutant mice. Congruent with this finding, CpG methylation at a site within the *Ryr2* gene was higher in TRα1 + m mice, with maternal T3 treatment reversing this. Notably, *Ryr2* is strongly linked to heart rate, as it encodes the ryanodine receptor 2, which governs the release of calcium from sarcoplasmic reticulum into the cytosol, triggering cardiac muscle contraction[37]. Mice with an inducible, conditional knockout which reduces cardiac *Ryr2* expression by 50% developed bradycardia[40], indicating that reduced expression of this gene is sufficient by itself to cause the observed cardiac phenotype. However, the exact contribution of this particular gene in the context of several other pathways that are also altered in TRα1 + m heart, remains to be established. Furthermore, additional studies to link methylation status of this specific CpG site in *Ryr2* to altered expression of this gene, are needed.

Whilst RTH due to TRβ mutations has long been recognized[41,42], individuals with a mutation in TRα1 were discovered relatively recently[15,43] and often exhibit resting bradycardia[16]. Our studies show that such reduction in their resting heart rate together with refractoriness to the expected rise following thyroxine therapy of patients in high dosage, is likely to be due to effects of mutant TRα1 during embryonic development, irreversibly reprogramming cardiac pacemaker function. These observations suggest that thyroxine therapy of RTHα, in dosage sufficient to overcome hormone resistance, can be undertaken without inducing the detrimental tachycardia associated with elevated circulating thyroid hormones. Nevertheless, as several genes (e.g., *Kcnh2*) that are downregulated in TRα1R384C mutant mice are associated with arrhythmia or sudden death[44], close monitoring of thyroxine-treated RTHα patients seems advisable.

Overall, our data show that impaired TRα1 signalling during embryonal development leads to cardiac reprogramming which confers resistance to T3-induced tachycardia. At the molecular level, such resistance occurs despite expected upregulation of *Hcn2* and *Hcn4*, indicating that these classical T3 target genes alone are not sufficient to mediate the tachycardia of hyperthyroidism. Reduced expression of other target genes (the *Ryr2* calcium channel; the *Kcnh2* potassium channel), likely contribute to the resistance to T3-induced tachycardia seen in TRα1 mutant mice. Most importantly, these observations provide a mechanistic understanding for the lack of tachycardia in thyroxine-treated RTHα patients, suggesting that high dose hormone therapy of this disorder may be safer than previously thought.

## Methods
### Human studies
Patients were born with a congenital defect. Their diagnosis of RTHα has been described previously[18], with their treatment with thyroxine being standard of clinical care and follow-up over 8 years part of an observational study. All investigations were undertaken as part of

ethically approved protocols (RTHα: Cambridgeshire Local Research Ethics Committee LREC 98/154; Thyrotoxicosis: REC 05/Q0108/117; Reference measurements in healthy subjects: REC 06/Q0108/84), with prior, written, informed consent of participants before studies according to CARE guidelines and in compliance with the Declaration of Helsinki principles. The authors affirm that human research participants or their parents/legal guardians provided written informed consent for publication of the potentially identifiable medical data included in this article. Thyroid hormone measurements (free thyroxine (FT4), free triiodothyronine (FT3), TSH) were measured by Advia Centaur (Siemens, Germany). Reverse T3 (RT3) was measured by Quest Diagnostics (San Juan Capistrano, USA) or by in-house assay. Resting energy expenditure (REE), measured by indirect calorimetry at baseline and following thyroxine therapy as described previously[17,18], was corrected for body composition and cross-calibrated to GE Lunar iDXA[45,46], and compared to a healthy adult reference population[47,48] to derive a standard deviation (or Z) score[17,18]. Actiheart (CamnTech, Fenstanton, Cambridgeshire, UK; software version 4.0.116 and 5.1.29), a coin-shaped device attached to the praecordium, which has been validated in human subjects[49], recorded heart rate and movement by accelerometry. In each patient (two male, one female), data from the first full 24 h period was analysed further. Heart rate data, collected in 15-s epochs and filtered to exclude missing and irregular data points, was averaged to hourly intervals for 24 h.

## Animals and in vivo recordings

Animals were housed in groups at 22 or $30 \pm 1\,°C$ as indicated at constant 12-h light/dark cycle at 40–60% humidity with ad libitum access to food (fortified diet #1314, Altromin, Germany) and water. Experiments were conducted in male mice at the age of three to 6 months. Wild-type C57Bl/6NCr were purchased from Charles River Laboratories (Charles River, Germany), TRα1R384C heterozygous mutant mice[12] and control littermates were bred on the C57Bl/6NCr background at the Gemeinsame Tierhaltung of the University of Lübeck. Hyperthyroidism was induced by treatment with 0.5 mg/L 3,3',5-Triiodo-L-thyronine (T6397, Sigma-Aldrich, Germany) in 0.01% BSA and tap water for 12 days or during the second half of pregnancy until the day before birth (embryonic day 12.5–17.5)[23]. For recording heart rate, body temperature, and activity in freely moving mice an implantable radiotelemetry system with transmitters and receivers (G2-HR E-Mitter and ER-4000 Receiver Plates with Telemetry Software Vital View 4.200.2; Philips Respironics, USA) was used[30]. To quantify autonomic input, the mice were injected with scopolamine methyl bromide (0.1 mg/kg body weight i.p.; Sigma-Aldrich, Germany) to block muscarinic receptors (PSNS) and 45 min later with timolol maleate (1 mg/kg body weight i.p.; Sigma-Aldrich, Germany) to block β-adrenergic receptors (SNS). The differences in heart rate were calculated[14] and the intrinsic heart rate was recorded after both blocks. Animals were monitored daily, euthanized using carbondioxide or isoflurane in combination with cervical dislocation, and all animal procedures were approved by the Ministerium für Energiewende, Landwirtschaft, Umwelt, Natur und Digitalisierung des Landes Schleswig-Holstein (MELUND, State Ethical Animal Welfare Committee, Germany).

## Intracerebroventricular T3 administration

For intracerebroventricular (i.c.v.) injection, a cannula was placed in the lateral ventricle of the mouse[50]. Then 8 ng T3 were infused in 2 μl volume (dissolved in saline with sodium hydroxide, then adjusted to pH 7.4). Controls received the same solution without T3.

## T3 and T4 serum concentrations

Serum concentrations were measured using commercially available ELISA kits for total T4 (EIA-1781; DRG Instruments GmbH, Germany) and total T3 (DNOV053; NovaTec Immundiagnostica GmbH, Germany).

## Norepinephrine turnover

Norepinephrine turnover was determined as described previously[51]. Briefly, mice were injected with the tyrosine hydroxylase inhibitor α-methyl-DL-tyrosine methyl ester hydrochloride (i.p. 120 mg/kg) (AMPT, M3281, Sigma-Aldrich, Germany) and sacrificed after 0, 2, 4 and 6 h, respectively. Catecholamines were extracted in 2% perchloric acid (PCA, 48–50%, 44464, Alfa Aesar, MA, USA) and quantified[52].

## Gene expression studies

RNA was isolated using QIAGEN RNeasy Kits (QIAGEN, Germany). Microarray analyses were conducted with Clariom S Mouse Gene Chips (Affymetrix, Germany; cDNA synthesis, hybridization and quality check at ATLAS Berlin, Germany). The data were analysed using the Transcriptome Analysis Console (TAC Version 4.01, Affymetrix, Thermo Fisher, Germany) and filtered for expressed genes with log >4. MATLAB (MathWorks R2022b, USA) was used to generate the principal component analysis, R with the respective package was used to generate the Venn Diagram and the GOplot. Genes involved in heart rate regulation (GO:0002027) and potassium channels were analysed using a two-way ANOVA for T3 and TRα1 as independent factors and corrected for multiple comparisons by controlling the false discovery rate using the Benjamini, Krieger and Yekutieli method with $q < 0.05$ considered significant. The top ten genes were obtained by sorting after smallest $P$ value. Genes that were subsequently significantly different at baseline and restored after T3 treatment (i.e., no longer significantly different in the same direction between wild-type and T3-treated TRα1 + m mice) were considered as bona fide TRα1 target genes. Genes that were significantly different at baseline and not restored after T3 treatment (i.e., still significantly different between control or T3-treated wild-type and T3-treated TRα1 + m mice) were considered permanently altered in TRα1 + m mice. The heat maps were generated using HeatMapper[53].

## DNA methylation analysis

DNA methylation was analysed with the Infinium Mouse Methylation BeadChip Kit (Illumina) using 500 ng DNA as input. Quality control, background correction, dye-bias correction and inter-array quantile normalization were performed after exclusion of outliers with the bioconductor package "ENmix" version 1.34.0[54]. For background correction, the out-of-band option was chosen, which uses the unused color channel to estimate the background intensity. The option RELIC (REgression on Logarithm of Internal Control probes) was used for dye-bias correction. Statistical analysis and visualization were performed in R-4.1.0 (The R-Foundation for Statistical Computing, Vienna, Austria) with the ggplot2 and pheatmap libraries. $P$ values were adjusted for multiple testing by controlling the FDR using the method of Benjamini–Hochberg.

## Statistical analysis and figure preparation

Data were analysed using GraphPad Prism 8/9 (GraphPad Software, San Diego, CA, USA) or Excel 2016/2010/365 Version 2303. If a two-way ANOVA was used, the results of the respective post-hoc tests are given in the figure. Additional information on the specific tests used as well as the $F$- and $P$ values can be found in Supplementary Table 1. Parts of the figure were drawn by using pictures from Servier Medical Art licensed under a Creative Commons Attribution 3.0 Unported License.

## Reporting summary

Further information on research design is available in the Nature Portfolio Reporting Summary linked to this article.

## Data availability
The array data have been deposited at GEO under GSE206337 (microarray data in Fig. 3), GSE227847 (microarray data in Fig. 4) and GSE226701 (methylation array in Fig. 4) and are freely available. The telemetry data have been deposited at Zenodo (https://zenodo.org/record/7961873) and are freely available. Source data are provided with this paper.

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

## Acknowledgements

We thank the Gemeinsame Tierhaltung Lübeck for excellent animal caretaking. Research in human participants is supported by the Wellcome Trust (Investigator Award 210755/Z/18/Z to K.C.) and NIHR Cambridge Biomedical Research Centre (to K.C., C.M., G.L.) and was conducted in the NIHR Cambridge Clinical Research Facility. S.H. and K.L.R. were also supported by the NIHR Cambridge Biomedical Research Centre (IS-BRC-1215-20014). We are grateful for funding from the German Research Council DFG in the framework of CRC/TR296 "LocoTact" (funding ID 424957847 to J.M.), GRK1957 "Adipocyte Brain Crosstalk" to J.M., MI1242/3-2 to J.M. and OE723/2-1 (funding ID 434396546 to R.O.). We also thank Prof. Ursula Ravens for constructive criticism on our manuscript.

## Author contributions

R.D., L.W., S.H., S.C.S., K.J., M.P., R.O., N.L.A., J.R., G.L. and C.M. undertook studies; R.D., L.W., S.H., C.K., S.C.S., C.G., K.J., M.P., R.O., K.L., K.R., A.A., H.K., K.C. and J.M. analysed the data; F.S., A.I., A.F., T.S.B., J.W.D. and A.A. provided crucial methods; R.D., R.O., C.M., A.A., K.C. and J.M. designed the experiments; R.D., K.C. and J.M. drafted the manuscript; all authors read, discussed and approved the manuscript.

## Funding

## Competing interests

The authors declare no competing interests.
