## [Peer Review File · Nature Communications]

REVIEWER COMMENTS

Reviewer #1 (Remarks to the Author):

In this study, authors analyzed the resistance to thyroid hormone due to A263V mutation in TRa1 on cardiac function in patients, and explored possible mechanisms by using a TRa1 mutant mice. The data are interesting, but data supporting the most exciting discovery, the developmental origin of abnormal gene expression, are very preliminary. The following are detailed comments:

1. Per the title, this study is mainly about the developmental defect preventing thyroid hormone induced tachycardia, which is also the most exciting discovery. However, there was only mRNA expression of 4 genes reported, which is far from sufficient. To address, 1) the authors need to analyze the T3 and T4 concentrations in maternal and fetuses due to maternal T3 injection, considering the presence of placental barrier; the thyroid hormone status of offspring mice should also be analyzed; 2) Cardiac structural and functional changes should be analyzed due to maternal T3 injection in both fetal and offspring heart; 3) Why were the mRNA expression of these genes changed? To address, a transcriptome profiling of both fetal and offspring mice is needed; 4) To have long-term impacts on gene expression, it could be due to cellular composition changes or DNA methylation changes. Data to support either these changes due to maternal T3 administration is needed; 5) Changes in PSNS and SNS in offspring mice should also be conducted.
2. For Figure 2E, related measurements should also be conducted in mutant mice.
3. Discussion section should include the developmental origins of changes, as well as potential mechanisms involved.
4. A conclusion section should be added.
5. The statistic section should include more details, such as Sidak's multiple comparison, which might not be able to do using Prism 9?

Reviewer #2 (Remarks to the Author):

This was a well done translational study examining a mutation in TRalpha and heart rate regulation. The mouse model appears to reflect the human condition well, especially in 30degreeC conditions. The study provides new insight suggesting that the HR changes are intrinsic to the myocardium rather than autonomic. The prenatal T3 treatment of the mutant mouse model showing reprogramming of the adult heart is clinically relevant and a very interesting result. Results also suggest that higher doses of T4 in adult humans with this condition may be safer than believed.

Reviewer #3 (Remarks to the Author):

The authors report that cardiac developmental defect prevents thyroid hormone induced tachycardia in the syndrome of Resistance to Thyroid Hormone alpha. Overall, the MS is quite convincing, and I only have a few suggestions for the authors.

The data clearly demonstrate that T3 exposure and resulting TH alpha activation (or lack thereof) during development results in a permanent change in cardiac ion channel expression. It seems likely that this is the result of differential methylation of the ion channel genes in the adult animals. The authors should provide experimental evidence to corroborate that hypothesis. Without that, the reader is left wondering about the mechanisms responsible for the observation.

The ECG records shown in Fig 3 are of such low quality that this reviewer is very uncomfortable making any inferences regarding QRS, PQ and QT duration. This figure should be dropped, or experiments repeated with high quality recording systems, which may require anesthesia. Evidently, the ECGenie Clinic system used by the authors is not capable of generating high-quality ECG records required for QRS or QT analysis, which is challenging in mice even with high fidelity records. If the authors are confident that the QRS duration is altered, this needs to be supported by cardiac conduction velocity measurements using optical mapping, and single cell measurements of Na currents. Since this figure is a side point to the overall story, it could be simply deleted.

The authors should show individual datapoints rather than bar graphs in their figures

Reviewer #1 (Remarks to the Author):

In this study, authors analyzed the resistance to thyroid hormone due to A263V mutation in TR α 1 on cardiac function in patients, and explored possible mechanisms by using a TR α 1 mutant mice. The data are interesting, but data supporting the most exciting discovery, the developmental origin of abnormal gene expression, are very preliminary.

We thank the reviewer for the constructive comments and suggestions, which helped to improve the manuscript substantially. We conducted all experiments that were possible within the timeframe of 3 months given for the revision, and we hope that the reviewer finds the manuscript now acceptable for publication. Please find the details below.

The following are detailed comments:

1. Per the title, this study is mainly about the developmental defect preventing thyroid hormone induced tachycardia, which is also the most exciting discovery. However, there was only mRNA expression of 4 genes reported, which is far from sufficient. To address, 1) the authors need to analyze the T3 and T4 concentrations in maternal and fetuses due to maternal T3 injection, considering the presence of placental barrier; the thyroid hormone status of offspring mice should also be analyzed; 2) Cardiac structural and functional changes should be analyzed due to maternal T3 injection in both fetal and offspring heart; 3) Why were the mRNA expression of these genes changed? To address, a transcriptome profiling of both fetal and offspring mice is needed; 4) To have long-term impacts on gene expression, it could be due to cellular composition changes or DNA methylation changes. Data to support either these changes due to maternal T3 administration is needed; 5) Changes in PSNS and SNS in offspring mice should also be conducted.

Answer: We thank the reviewer for these helpful suggestions and giving us the opportunity to provide additional experimental data to support our conclusions. In response to these comments, we have now analyzed the T3 and T4 levels in the pregnant dams, their embryos and the offspring (point 1).

The data show the expected significant increase in serum T3 in the treated mothers and their embryos at GD17, and no effect of the maternal treatment on either hormone in the adult offspring of both genotypes. The data have been included as Suppl Fig 4A.

While we could not study structural changes (point 2) or autonomic innervation (point 5) in the offspring hearts due to the restricted time-frame of the revision, we followed the reviewers advice to conduct a transcriptome profiling of the offspring (point 3) - a suggestion that we are particularly grateful for! This approach turned out to be extremely valuable, as we identified several candidates that could explain the tachycardia resistance in the TR α 1+m mice, especially *Ryr2*, which we missed in our limited PCR study. The transcriptome data are now included in Fig 4A-C and Suppl Fig 4B:

Figure 4

Furthermore, we also analyzed DNA methylation changes as suggested (point 4) using Illumina Bead Mouse Methylation chips. The data revealed several clusters of differentially methylated CpGs that are different in the TR α 1+m hearts and indeed restored by the maternal T3 treatment to wildtype levels. The methylation data have been included in Fig 4D and Suppl Fig 4.

2. For Figure 2E, related measurements should also be conducted in mutant mice.

Answer: This is an important suggestion raised by the reviewer. We have now repeated the i.c.v. study in TR α 1+m mice and recorded heart rate for 6 hours after injection, SNS/PSNS activity and locomotion, showing no difference between T3 i.c.v. and control solution, comparable to what observed in the wildtype animals. The data have been included in the revised manuscript as Fig 2F, and Suppl Fig 2C and E.

3. Discussion section should include the developmental origins of changes, as well as potential mechanisms involved.

Answer: We have included the respective section into the discussion and elaborated on the potential mechanisms through fetal programming by CpG methylation based on our genomewide DNA methylation data.

4. A conclusion section should be added.

Answer: We have added a conclusion section to the discussion.

5. The statistic section should include more details, such as Sidak's multiple comparison, which might not be able to do using Prism 9?

Answer: We apologize for the missing information. We have used GraphPad Prism 7-9 simultaneously, and found no differences in p-values between the two versions. The statistical details for every analysis are provided in Suppl Table 1, including the used test and the respective F and p-values.

Reviewer #2 (Remarks to the Author):

This was a well done translational study examining a mutation in TRalpha and heart rate regulation. The mouse model appears to reflect the human condition well, especially in 30degreeC conditions. The study provides new insight suggesting that the HR changes are intrinsic to the myocardium rather than autonomics. The prenatal T3 treatment of the mutant mouse model showing reprogramming of the adult heart is clinically relevant and a very interesting result. Results also suggest that higher doses of T4 in adult humans with this condition may be safer than believed.

Answer: We thank the reviewer for the positive and encouraging comment on our manuscript.

Reviewer #3 (Remarks to the Author):

The authors report that cardiac developmental defect prevents thyroid hormone induced tachycardia in the syndrome of Resistance to Thyroid Hormone alpha. Overall, the MS is quite convincing, and I only have a few suggestions for the authors.

Answer: We thank the reviewer for the kind and encouraging comments. Please find the detailed answers to your comments below.

The data clearly demonstrate that T3 exposure and resulting TH alpha activation (or lack thereof) during development results in a permanent change in cardiac ion channel expression. It seems likely that this is the result of differential methylation of the ion channel genes in the adult animals. The authors should provide experimental evidence to corroborate that hypothesis. Without that, the reader is left wondering about the mechanisms responsible for the observation.

Answer: We appreciate the excellent suggestion of the reviewer and have now analyzed differential CpG methylation in hearts of wildtype and TR α 1+m mutants with or without T3 treatment during the second half of pregnancy using the Illumina Infinium Bead mouse chip. The results clearly revealed an effect of TR α 1 and maternal T3 on overall DNA methylation (Fig 4E) and even more importantly identified distinct clusters of differentially methylated CpG sites that were altered in the TR α 1+m mutant hearts and restored to wildtype levels by maternal T3 treatment (Fig 4G).

Most remarkably, by combining all our transcriptome data with methylome data, we were able to identify the calcium channel Ryr2 as most promising candidate gene, as its expression was a) different in adult TR α 1+m mice, b) not restored by adult T3 treatment, c) improved by maternal T3 treatment, and d) exhibited a CpG site in the gene body which mirrored the methylation reversal by maternal T3

treatment (Fig 4H). Given that an inducible adult Ryr2 knockout causes bradycardia in mice, thus demonstrating that an altered expression of this single gene can cause the observed phenotype, this candidate provides a possible mechanism for the developmentally programmed cardiac resistance to T3 in TR α 1+m mice.

The ECG records shown in Fig 3 are of such low quality that this reviewer is very uncomfortable making any inferences regarding QRS, PQ and QT duration. This figure should be dropped, or experiments repeated with high quality recording systems, which may require anesthesia. Evidently, the ECGenie Clinic system used by the authors is not capable of generating high-quality ECG records required for QRS or QT analysis, which is challenging in mice even with high fidelity records. If the authors are confident that the QRS duration is altered, this needs to be supported by cardiac conduction velocity measurements using optical mapping, and single cell measurements of Na currents. Since this figure is a side point to the overall story, it could be simply deleted.

Answer: We truly appreciate the expert insight of the reviewer on the ECG data. The reason why we used the ECGenie system over the classic Powerlab setup was that it could easily be added to our existing animal licenses as it is entirely non-invasive, and secondly that it works with non-anaesthetized animals thus eliminating isoflurane as possible covariate on the autonomic regulation. However, the reviewer is correct that the sampling frequency is limited to 2kHz with the ECGenie system. To avoid the problems raised by the reviewer we have decided to eliminate Figure 3 as suggested.

The authors should show individual datapoints rather than bar graphs in their figures

Answer: We appreciate this suggestion and have added the individual data points to all bar graphs.

REVIEWERS' COMMENTS

Reviewer #1 (Remarks to the Author):

The authors addressed my concerns.

Reviewer #3 (Remarks to the Author):

The authors have addressed all of my concerns satisfactorily. Outstanding work!

REVIEWERS' COMMENTS

Reviewer #1 (Remarks to the Author):

The authors addressed my concerns.

Reviewer #3 (Remarks to the Author):

The authors have addressed all of my concerns satisfactorily. Outstanding work!

A: We thank the reviewers for their kind and constructive comments.